
# Real-time *ab initio* description of the photon-echo mechanisms in extended systems: the case study of bulk GaAs

**Marco D'Alessandro[1][*] and Davide Sangalli[2]**

**1** Istituto di Struttura della Materia-CNR (ISM-CNR) , Division of Ultrafast Processes in Materials (FLASHit), Via del Fosso del Cavaliere 100, 00133 Roma, Italy
**2** Istituto di Struttura della Materia-CNR (ISM-CNR), Division of Ultrafast Processes in Materials (FLASHit), Area della Ricerca di Roma 1, Monterotondo Scalo, Italy

[*] marco.dalessandro@ism.cnr.it

## Abstract

In this paper we present an *ab initio* real-time analysis of free polarization decay and photon echo in extended systems. As a prototype material, we study bulk GaAs driven by ultra-short laser pulses with a FWHM of $10\,\mathrm{fs}$. We compute the electronic polarization $P(t)$ and define a computational procedure to extract the echo signal. Results are obtained in both the low and high field regime, and compared with a two-levels system (TLS) model, with parameters extracted from the *ab initio* simulations. An optimal agreement with the TLS is found in the low-field case, whereas some differences are observed in the high-field regime where the multi-band nature of GaAs becomes relevant.

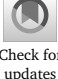

# 1  Introduction

The coherent control of the electronic polarization $\mathbf{P}(t)$ via ultra–short laser pulses and non-linear effects provides a powerful tool to study fundamental processes in semiconductors, such as dephasing time of complex excitations, and opens the opportunity to a wide range of applications for quantum information protocols [1–4] and quantum devices [5]. The photon-echo technique is the key mechanism behind many protocols in coherent spectroscopy [6–8]. Thanks to the development of ultra-short laser pulse, echo experiments can be nowadays used to explore ultra–fast electron dynamics in complex materials [9–11].

The principles upon which photon-echo relies are the same of the spin–echo technique, introduced more than 70 years ago to eliminate the so called "free induction decay" (FID) [12, 13], or quantum mechanical dephasing. A perturbing laser pulse acting on magnetic materials induces a dynamics, or spin-precession, of the macroscopic magnetization which however usually decay on a very short time–scale. Quantum mechanically, spin precession results from coherent magnon states oscillating at one or few well defined energies (or frequencies) $\omega_\lambda(\mathbf{q})$ with fixed transferred momentum (usually $\mathbf{q} = 0$ since the dynamics is optically activated). However, the existence of defects and impurities leads to an "inhomogeneous broadening", with a high number of similar frequencies $\omega_{\lambda,i}(\mathbf{q})$. The spin–precession involves a super-position of all such frequencies, and the global dynamics experiences a fast FID on a very short time scale ($T_2^*$), since all contributions quickly get out of phase. The same process can exist also in ideally perfect crystals if, for example, one considers the magnetization due to spin–carriers optically injected in the conduction band of a semi–conductors. In such case the coherent dynamics is associated to spin-flip transitions involving states in the continuum $\omega_{nm\mathbf{k}}(\mathbf{q}) = (\epsilon_{n\mathbf{k},\sigma} - \epsilon_{m\mathbf{k}-\mathbf{q},-\sigma})$ and FID happens on an even shorter time scale [1].

The key idea behind the echo technique is to use a second pulse, called echo pulse or rephasing pulse, to reverse the microscopic dynamics and bring back in phase all coherent terms. Doing so makes it possible to explore and exploit the coherence time ($T_2$) of the under-lying quantum states. $T_2$ can be substantially longer than $T_2^*$. Spin-echo is usually achieved by means of the so called $\pi$-pulses. Here $\pi$ (or fractions of $\pi$) refers to the pulse area [14,15], *i.e.* the time integral of the Rabi coupling $\Omega(t) = \mathbf{E}(t) \cdot \hat{\boldsymbol{\mu}}$. $\pi$-pulses lead to a global inversion of the microscopic dynamics due to strongly non linear effects, as can be understood in terms of the Rabi cycle in a two-levels system (TLS). The idea of spin-echo was followed by a very large number of developments, both for technological applications and fundamental investigation [16–18].

In the photon–echo case the macroscopic polarization $\mathbf{P}(t)$ takes the place of the magneti-zation, and the rephasing pulse is used to revert the "free polarization decay" (FPD) [19–24]. Photon–echo was first explored in isolated systems and quantum dots; more recently to study excitonic resonances and inter–band transitions in semi–conductors, where experiments have focused on the use of low intensity pulses. One of the reasons is that pulses with an area close to $\pi$ lead to band inversion, at least in some regions of the reciprocal space, which could result in a band collapse and a melting of the material [25]. At low intensity the echo is not realized for the whole signal, but one needs to expand the induced polarization $\mathbf{P}(t)$ in powers of the external fields and select the response at a specific order [26]. Experimentally this is achieved using a non collinear geometry between the perturbing and the echo pulse. In four–wave mixing (FWM) experiments for example the third order polarization, $\mathbf{P}^{(3)}$ is considered. The echo in $\mathbf{P}^{(3)}$ emerges as a consequence of the quantum dynamics generated by the echo pulse acting on the system at time $t = t_0 + \tau$, where the initial time $t_0$ is defined by the action of the perturbing pulse. Setting $t_0 = 0$, the general solution has re-phasing terms [10] of the form

---

[1]The energy dispersion due to the crystal momentum $\mathbf{k}$ takes the place of high number of poles $\omega_i$ due to defects and impurities.

$cos(\omega_{ij\mathbf{k}}(t-2\tau))$ and $sin(\omega_{ij\mathbf{k}}(t-2\tau))$. Clearly at time $t = 2\tau$ all terms get back in phase, and an echo signal is expected. The Rabi cycle driven by $\pi$–pulses, can be understood as a special case where the rephasing terms dominate the signal.

The modelling of echo experiments can be described by propagating the equation of motion for a set of uncoupled TLSs, usually within the rotating–wave approximation (RWA). In the simplest description, an isolated pole at fixed Rabi coupling is considered, with an energy dispersion due to some inhomogeneous broadening effect. Different works have gone beyond such approach, especially focusing on extended systems. Some papers considered realistic energy dispersion and the role of many-body interactions within the framework of the semiconductor Block equations and non–equilibrium Green function approaches [23, 26, 27]. Other focused on the effect of intra-band transions, within the two–bands model [28, 29]. On the other hand, while the popularity of *ab initio* real-time simulation is growing, with particular focus on the coherent electron and spin dynamics, little is known on the possibility of capturing *ab initio* free induction or polarization decay and subsequently echo signals. In a recent work [30] we have shown how the free induction decay of the magnetization can be modelled only with a very dense sampling of the Brillouin zone.

In the present work we aim at showing how the physics of echo mechanisms can be described in the theoretical framework of the real-time *ab initio* simulations. We use bulk GaAs as a prototype material and consider a laser pulse tuned resonant with the continuum, *i.e.* above the electronic gap of GaAs. The energy dispersion of the transitions $\omega_{ij\mathbf{k}}$, where the indexes $i$ and $j$ run on multiple bands, is accounted for by computing *ab initio* the electronic band structure on a very dense $\mathbf{k}$-mesh. Also the corresponding electronic dipoles are computed *ab initio* leading to a dense distribution of Rabi couplings $\Omega_{ij\mathbf{k}} = \mathbf{E} \cdot \boldsymbol{\mu}_{ij\mathbf{k}}$. We propagate the equation of motion (EOM) for the one-body density matrix of the system, under the action of both the perturbing and the rephasing laser pulses, beyond the RWA. We study the decay time of the polarization using a set of echo pulses with different time shifts, both in the low and in the high intensity regimes. The main goal of the present analysis is to have a *proof of concept* which demonstrates how the free polarization decay and the echo mechanism can be captured fully *ab initio*. In doing so we also relax some of the approximations introduced in the TLS and explore their effect.

The paper is organized as follows. In section 2 we present the fundamental computational elements used provide the *ab initio* description the GaAs and its coupling with the optical field responsible for the non-equilibrium dynamics. Section 3 contains the analysis of the echo mechanism in the low field regime, together with a comparison to the physics of the TLS, while results in the high intensity limit are presented in section 4. Lastly, a concise but self-consistent review of the TLS related equations, with particular emphasis on optical absorption and echo mechanism, is provided in appendix B.

## 2 Computational Methods

The equilibrium (quasi)-particle band structure $\epsilon_{n\mathbf{k}}$ of GaAs is computed using density functional theory (DFT) plus a scissor operator correction [31–33]. The value of the scissor is 0.955 eV and is chosen to match the experimental band gap of 1.42 eV. The DFT calculation is performed in the plane-wave basis using the Perdew-Burke-Ernzerhof (PBE) [34] exchange and correlation functional, as implemented in the Quantum ESPRESSO package [35, 36]. Scalar relativistic norm conserving pseudopotential has been used for both the Ga and As core electrons and the spin-orbit coupling has not been considered, since it provides negligible corrections to the observable electronic polarization. The total energy of the ground state and the band gap are converged using a $6 \times 6 \times 6$ Monkhorst-Pack $\mathbf{k}$ grid and an energy cutoff of

60 Ry. The equilibrium lattice parameter computed by volume relaxation is $a_{lat} = 5.59$ Å, in good agreement with previous calculations. The band structure along a high symmetry path in the Brillouin zone (BZ) is shown in the top panel of Fig. 1.

We then perform real-time electron dynamics simulations, in presence of two ultra–fast linear polarized monochromatic electric pulses, with frequencies tuned at $\omega_0 = 1.58$ eV, modulated by Gaussian envelopes:

$$\mathbf{E} = \mathbf{e}_x sin(\omega_0 t)\big(E_p f_p(t) + \alpha E_e f_e(t-\tau)\big). \tag{1}$$

Here $\mathbf{e}_x$ is the polarization versor, $E_{p,e}$ determine the amplitude of the two pulses, $f_{p,e}(t)$ describe the profiles of the Gaussian envelopes, and $\alpha$ is a phase factor. In what follows the first addend of (1) is the *pump* or perturbing field, whereas the second term is the *echo* or rephasing field. The temporal width of the two pulses is determined by their Gaussian envelopes. We use very short pulses, with a full width at half maximum (FWMH) $\sigma = 10$ fs. This corresponds to an energy spread $\Delta = 0.4$ eV around $\omega_0$. Note that the pump *starts* at $t = 0$, when the simulation begins, and it reaches its maximum intensity at $t_0 \simeq 12$ fs, whereas the echo is activated with a time delay $\tau$ and reaches its maximum intensity at $t_1 = t_0 + \tau$. In this analysis we consider values of $\tau \gg \sigma$, so the pump and echo fields do not overlap in time.

The non-equilibrium dynamics at the one-body level is codified in the EOM for the time-dependent one-body density matrix (DM) $\hat{\rho}(t)$. In the present work we consider a regime where free electrons and holes are generated, since the energy of the pulses of 1.58 eV is around $\approx 0.16$ eV above the GaAs gap. This rules out the generation of bound excitons and justifies the use of a time-dependent independent-particles TD-IP approximation. Possible scattering and relaxation mechanisms, which in turn produce a dephasing of the electronic degrees of freedom are accounted for in an effective way by adding a damping factor in the the the non-diagonal elements of the DM. This term is responsible for the emergence of an *irreversible* dephasing time, denoted as $T_2$, that affects the time-dependency of the observables.

The resulting EOM, expressed in the Kohn-Sham (KS) basis reads

$$i\partial_t \rho_{nm\mathbf{k}} = \omega_{nm\mathbf{k}} \rho_{nm\mathbf{k}} - \big[\hat{H}_I, \hat{\rho}\big]_{nm\mathbf{k}} - \eta_{nm\mathbf{k}} \rho_{nm\mathbf{k}}. \tag{2}$$

The interaction Hamiltonian couples the system to the perturbing field trough the dipole interaction, so $\hat{H}_I = -\mathbf{E} \cdot \hat{\boldsymbol{\mu}}$. The damping factor $\eta_{nm\mathbf{k}}$ is set to $1/T_2$ for conduction-valence (cv) indexes and zero otherwise. Accordingly only the *cv* sector of the DM is affected by the dephasing. We set $T_2 = 250$ fs, which implies that the amplitude of the coherent oscillations of $\rho_{nm\mathbf{k}}$ is reduced by a factor $\approx 10$ in 600 fs.

Eq. (2) is solved in the time domain by using the Yambo package [37, 38] that propagates the one-body DM through a second-order Runge-Kutta integrator. From the DM we can compute the time-dependent electronic polarization

$$\mathbf{P}(t) = \frac{1}{V_l} tr(\hat{\rho}(t)\hat{\boldsymbol{\mu}}), \tag{3}$$

where $V_l$ is the volume of the (direct) lattice. In what follows we look at the $x$ component of the polarization since the perturbing field is polarized in this direction. To simplify the notation we use $P_x = P$.

Eq. (3) includes the average over the sampling of $\mathbf{k}$-points used to compute the matrix elements $\rho_{nm\mathbf{k}}(t)$. Here we have identified the *optical active region*, denoted as $\mathbf{K}_{\text{pump}}$, as the subset of the whole BZ which includes transitions with energies in the range $[\omega - \Delta/2, \omega + \Delta/2]$, where $\Delta$ is the energy spread of the pulses. Only transitions belonging to $\mathbf{K}_{\text{pump}}$ give a relevant contribution to (3) and it is thus possible to provide a reliable evaluation of $P$, by using a very fine sampling in the $\mathbf{K}_{\text{pump}}$ region only, within a manageable computational effort.

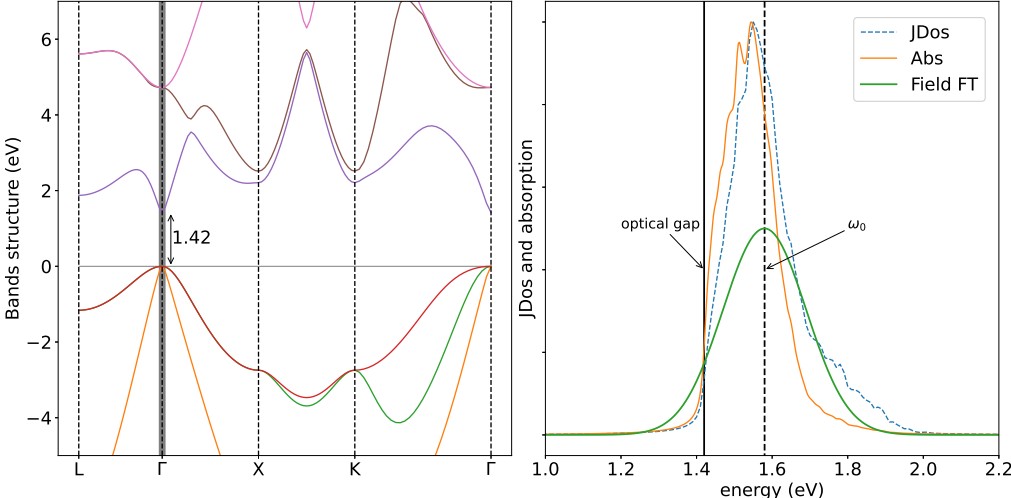

Figure 1: (Left) Quasi-particle band structure of GaAs. The grey thick line represents the size of the optical active region $\mathbf{K}_{\text{pump}}$, where an ultra fine sampling of the BZ is performed. (Right) Absorption spectrum and JDos of the optical transitions in the $\mathbf{K}_{\text{pump}}$ region. The green line shows the energy profile of the pulse.

The extension of $\mathbf{K}_{\text{pump}}$ is shown as a grey shaded area in the left panel of Fig. 1. $\mathbf{K}_{\text{pump}}$ is identified as a small cube of linear size $6 \times 10^{-2}$ (in units of $2\pi/a_{lat}$) centred at the $\Gamma$ point. Inside this volume a sampling of 11985 random points is performed, generated expanding by symmetry an initial set of 1000 random points uniformly distributed in the $\mathbf{K}_{\text{pump}}$ region. The JDos of the optical transitions and the associated (linear-response) absorption spectrum are reported in the right panel of Fig. 1. The figure also displays the energetic profile of the pulse, evidencing that the extension of $\mathbf{K}_{\text{pump}}$ is adequate to capture all the optically relevant transitions.

## 3 Photon echo at low intensity

We start by analyzing the time evolution of $P(t)$ due to a single pulse perturbing field in the weak field regime. According to the notation of (1) we set $E_p = 1 \times 10^4\,\text{kW/cm}^2$, with an associated fluence of $37.6\,\text{nJ/cm}^2$, and $E_e = 0$.

The time profile of $P(t)$ is shown in the top panel of starting from $t = 0$, when the pump is activated Fig. 2 (blue continuous line). It is characterized by fast oscillations convoluted with a rapid rise, dictated by the time profile of the pump (FWHM 10 fs), immediately followed by rapid decay. The time decay expected from the value of the dephasing time $T_2$ is also shown (blue dashed line), to underline that the decay of the real signal is much faster. This is due to the free polarization decay (FPD) mechanism. FPD emerges from the destructive interference of the individual $\mathbf{k}$-points contribution to $\mathbf{P}(t)$ in (3) [9, 11]. The FPD decay time can be estimated as $T_2^* = h/\delta_M$, where $h$ is the Planck constant and $\delta_M$ is the maximum value of the detuning in the $\mathbf{K}_{\text{pump}}$ region [39], i.e. the difference between the laser pulse frequency and the allowed optical transition energy. $\delta_M$ is roughly twice the energy spread $\Delta$ of the pump pulse, from which $T_2^* = 20\,\text{fs}$, so the FPD is much shorter than the physical decay $T_2 = 250\,\text{fs}$. A Gaussian fit of the polarization envelope (black continuous) allows us to extract quantitative information on $P(t)$. In particular, the maximum of $P(t)$ is located at $\approx 20\,\text{fs}$, i.e. 7 fs after the peak of the pump. The FWHM extracted from the symmetric fit of $\approx 16\,\text{fs}$ is an average

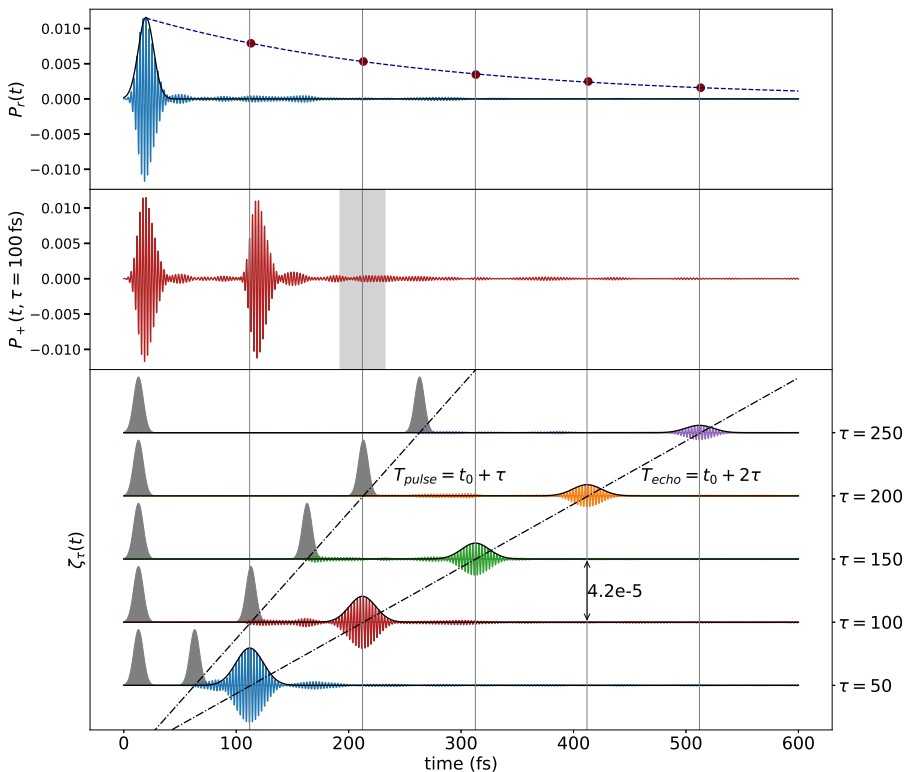

Figure 2: Real-time behavior of the *ab initio* polarization and echo signals. (Top) The blue curve represents the *reference* polarization due to the action of the pump field only. The blue-dashed line describes the time decay of the signal as expected from the physical dephasing time $T_2 = 250$ fs. The black continuous line shows the Gaussian fit of the real-time signal, its temporal width denotes an effective decay constant of 20 fs in agreement with the FPD expected dephasing time. The red dots represent the integrated intensity of the echo signals of the bottom panel. The plot evidences that these signals scale in agreement with the $T_2$ dephasing time and are not affected by the FPD mechanism. (Middle) $P_+$ signal due to the combined action of the pump and the echo fields, with a relative time delay $\tau = 100$ fs. The grey-shaded area evidences the temporal region in which the echo signal is expected to be generated. (Bottom) Echo signals, extracted according to the formula (4), for various values of the delay $\tau$ (specified in the $y$-axis on the right side of the panel). Echos are displayed as colored curves while the profile of the pump and echo fields are plotted as a gray shaded areas. The vertical arrow shows the order of magnitude the echo signals.

between the rise and the FPD. The energy spread determines both the rise, via the time profile of the Gaussian pulse, and the subsequent decay via the FPD mechanism. It is then natural that the two times are similar. In particular, since the initial frequencies of the pump are filtered by the frequency dependent polarizability, $\alpha(\omega)$ determines the relative duration of the two. For $\alpha(\omega)$ peaked at $\omega_0$, $T_2^*$ is longer (as in the present case). For $\alpha(\omega)$ with a deep at $\omega_0$ $T_2^*$ would be shorter. This, however, has nothing to do with the physical decay time which can be investigated via the echo technique.

The real-time dynamics of the polarization $P(t)$ changes when also the echo field is considered. Here we use an echo pulse with the same amplitude of the pump, $E_p = E_e$. In order to extract the echo signal we perform three distinct simulations. The first one, called *reference*,

includes only the pump, and the associated polarization is denoted as $P_r$. The second one includes also the echo field with positive phase factor $\alpha = 1$, and the associated polarization is denoted as $P_+$. In the third one the echo field is taken with negative phase factor $\alpha = -1$, and produces the polarization $P_-$.

The polarization $P_+$ associated to an echo delay $\tau = 100\,\text{fs}$ is shown in Fig. 2 (middle panel). The two observed peaks, very similar in amplitude and shape, are due to coupling of the system with the pump and echo fields. However, no signal is visible around $t = 2\tau$, where the echo signal should emerge. To extract the echo signal from the real-time dynamics of the polarization, we have defined

$$\zeta_\tau(t) = \frac{1}{2}\big(P_+(t,\tau) + P_-(t,\tau)\big) - P_r(t). \tag{4}$$

We observe that the leading order terms in the polarization, *i.e.* the two Gaussian peaks that were visible in the time profile of $P_+$ (Fig. 2, middle panel), cancel in the construction of the $\zeta_\tau(t)$ signal and the echo peak clearly emerges (Fig. 2, bottom panel). Notice, however, that the scale of the $\zeta_\tau(t)$ signal is about $1 \times 10^{-3}$ of the amplitude of the complete polarization $P_+$, denoting that the extraction of the echo signal is realized with a low efficiency. The time profile of $\zeta_\tau(t)$ is fully symmetric. The maximum of the amplitude is located at $2\tau + t_0$. The value is extracted from a Gaussian fit of the amplitude profile (black continuous line). So the echo signal emerges exactly with a delay $\tau$ after the action of the echo pulse. The analysis of the shape of the Gaussians also evidences that echo curves are larger than the $P_r$ signal. The FWHM of the echo envelopes is longer that the FPD, in between 27.7 fs and 30 fs depending on the dime delay, and with no evident dependence on $\tau$.

$\zeta_\tau(t)$ allows us to provide a FPD-free retrieval of the polarization generated by the pump field. To clarify this point, we have computed the areas of the Gaussian profile of the reference polarization ($A_r$) and of the echo signals for different time delays ($A_e(\tau)$). These quantities express an estimate of the integrated intensity of the associated signal and are reported in the top panel of Fig. 2 (filled dots). Their time behavior follows the low

$$A_e(\tau) = A_r \lambda e^{-2\tau/T_2}, \tag{5}$$

where $\lambda = 5.5 \times 10^{-3}$ represents the efficiency of the echo signal retrieval. Thus the echo signal scales according to $e^{-2t/T_2}$ and is free from the FPD decay as expected. The same time behavior of Eq. (5) is achieved if the amplitude of the echo signal is considered, instead of its area, and in this latter case the estimated efficiency is $\lambda' = 2 \times 10^{-3}$.

## 3.1 Perturbative analysis

We perform a perturbative analysis to better clarify the physical meaning of the equation (4). The observable polarization $P$ induced by a perturbing field can be expanded in powers of the field amplitude as follows

$$P(t) = P^{(0)}(t) + P^{(1)}(t) + P^{(2)}(t) + \cdots. \tag{6}$$

$P^{(0)}(t)$ does not depend on the perturbing field and it is zero (or constant for ferro-electric materials) if the laser pulse acts on the material in its ground state (GS). Here we include this term since we need to analyze the response to the echo field that acts onto the material out of equilibrium. We can now perform the perturbative expansion of the $P_\pm(t)$ in powers of the echo pulse. According to the analysis developed in Appendix A we obtain

$$P_\pm(t,\tau) = P_r(t) \pm \sum_{i \in \text{odd}} P^{(i)}(t,\tau) + \sum_{i \in \text{even}} P^{(i)}(t,\tau). \tag{7}$$

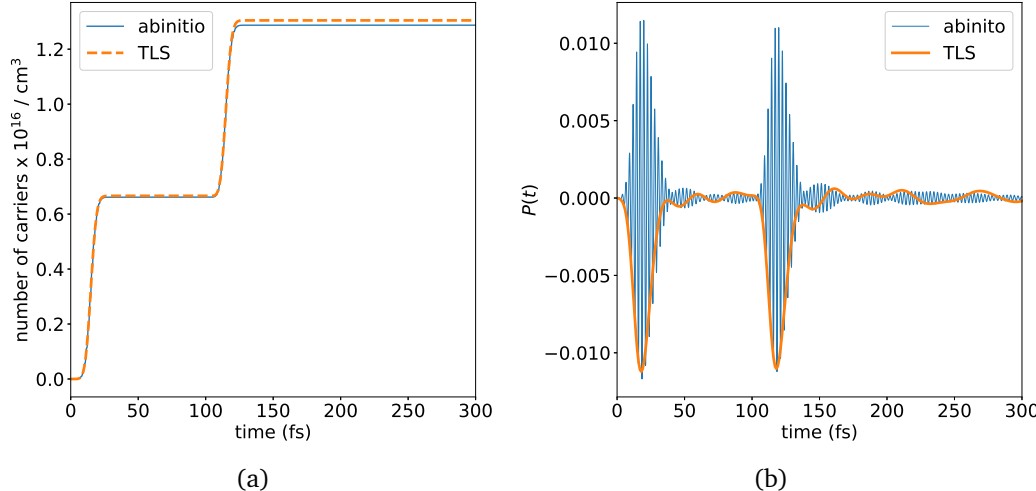

Figure 3: (a) Time dynamics of the carriers in the low-field regime. The blue (continuous) line shows the *ab initio* real-time simulation and the orange dotted line is result of the TLS modelization of the system. (b) Time dynamics of the polarization in the low-field regime. The blue and orange curves show the *ab initio* real-time and the TLS results, respectively. Note that TLS results are computed in the *rotating frame*, so the oscillations of frequency $\omega_0$ are absent.

Here the zero-th order term is the reference polarization due to the pump. The odd order coefficients of the expansion are opposite for the $\alpha = \pm 1$ cases, while the even order ones do not depend on the value of the phase factor.

Plugging this expansion in (4) provides

$$\zeta_\tau(t) = \sum_{i \in \text{even}} P^{(i)}(t, \tau). \tag{8}$$

The leading term in $\zeta_\tau(t)$ is the quadratic contribution in the echo field while the dependence on the pump is included implicit (and in a non-perturbative fashion) since it appears as the boundary condition of the perturbative equations of the density matrix. For weak pump and echo fields the $P^{(2)}(t, \tau)$ term can be further expanded in the pump amplitude and the resulting leading contribution evidences a linear dependence on the pump field. In this limit we recognize that the physical content of (8) is equivalent to the cubic term $P^{(3)}$, which is the observable typically considered in the FWM experiments.

## 3.2 Comparison with the physics of two-levels system

The results presented in this section are obtained in the low-pumping regime since the actual fluence of the pulses is able to excite only a small fraction of charge in the conduction bands and the transition resonant with single photon excitation are highly dominant. This is confirmed by the analysis of the blue curve of Fig. 3a which shows the time-dependent growth of the number of carriers, *i.e.* the **k**-points averaged number of electrons in the (first) conduction band in each cell. The shape of the curve, with a double plateau and two fast growth ramp in correspondence of the pump and echo pulses, follows exactly the profile of the fluence of the pulses, denoting that the absorption is described by a linear response process.

In this regime the real-time dynamics, that in general involves multi-bands transitions between occupied and empty states, can be reproduced using a strong simplification based on the physics of the two-levels systems (TLS). At the TLS level the non-equilibrium dynamics is

governed by the EOM

$$\dot{p} = -i\delta p - \frac{1}{2}|\Omega|I - \eta p \,,$$
$$\dot{I} = |\Omega|(p + p^*) \,, \tag{9}$$

which describe the time evolution of the two-dimensional density matrix parametrized in terms of the *inversion I* and *polarization* variables

$$p = e^{-i\omega_0 t}\rho_{12} \,, \quad I = \rho_{11} - \rho_{22} \,, \tag{10}$$

and $|\Omega|$ is a (positive) time-dependent coupling among $p$ and $I$. A detailed description of the notation and assumption adopted to describe the physics of a TLS and its coupling with the perturbing field is presented in Appendix B.

In this framework each optical transition is treated as a single TLS, giving rise to an ensemble of independent systems, and the observables are built as an average over the ensemble. We include all the optical transitions with energy inside the energy spread $\Delta$ of the pulses. Accordingly, each **k**-point can give rise to zero, one or several TLS depending on the local band energy structure, for a total of $N$ systems. In the present, case the 11895 points of the actual sampling of the $\mathbf{K}_{\text{pump}}$ region correspond to 34875 allowed transition, and thus to the same number of independent TLS. Each TLS is characterized by its proper value of the detuning and dipole matrix element, which in turn determines the values of the Rabi couplings $\Omega_p$ and $\Omega_e$, associated to the pump and echo fields, respectively. All the relevant parameters are inherited from the *ab initio* analysis, so a direct comparison between the results of the two approaches can be performed. To this scope we have developed a python tool [40] to manage the numerical solution of the EOM (9), for all the $p$ and $I$ variables.

The number of carriers $n_{\text{TLS}}$ and the observable polarization $P_{\text{TLS}}$ are computed as average of the equations (25) and (26) over the TLS ensemble. The results, corresponding to a time delay $\tau = 100\,\text{fs}$, are presented in Fig. 3a and Fig. 3b, where a comparison with the same quantities computed in the *ab initio* framework is also performed. We observe an excellent agreement of the two approaches, which confirms the validity of the approximations described above, for this fields intensity regime. Note that the $P_{\text{TLS}}$ curve reproduces the envelope of the *ab initio* polarization since the TLS solution is built in the rotating-frame, where the *fast* oscillations of frequency $\omega_0$ are absent.

Further insight on the physical mechanisms behind the emergence of the echo signal, as defined in (4), can be comprehended trough the perturbative analysis of the polarization variable in a TLS. In particular, the perturbative expansion of the EOM (9), truncated respectively to the first and to the second order in the pulse and echo fields, provide a general structure of the polarization of the form

$$p(t) = \theta_p p^{(1,0)} + \theta_e p^{(0,1)} + \theta_p \theta_e^2 p^{(1,2)} \,, \tag{11}$$

where we have introduced the apex notation $p^{(i,j)}$ to indicate the $i$-th and $j$-th orders solution in the pump and echo fields. Here the first and the second addends describe the linear effects of the pump and echo fields, respectively. The last one, which actually is the responsible of the echo signal, is the cubic term where the effects of both the fields are coupled. Under this perspective, the effect of the formula (4) is to remove the linear terms in the expansion (11).

A derivation of equation (11) is discussed in Appendix B, where simple formulas for the coefficients of the expansion have been derived for square-shaped pulse and small detunings. Within these approximations the coefficients read

$$p^{(1,0)} = -\frac{e^{-\eta t}}{2}e^{-i\delta t} \,, \quad p^{(0,1)} = -\frac{e^{-\eta(t-\tau)}}{2}e^{-i\delta(t-\tau)} \,,$$
$$p^{(1,2)} = \frac{e^{-\eta t}}{8}\left(e^{-i\delta t} + e^{-i\delta(t-2\tau)}\right) \,. \tag{12}$$

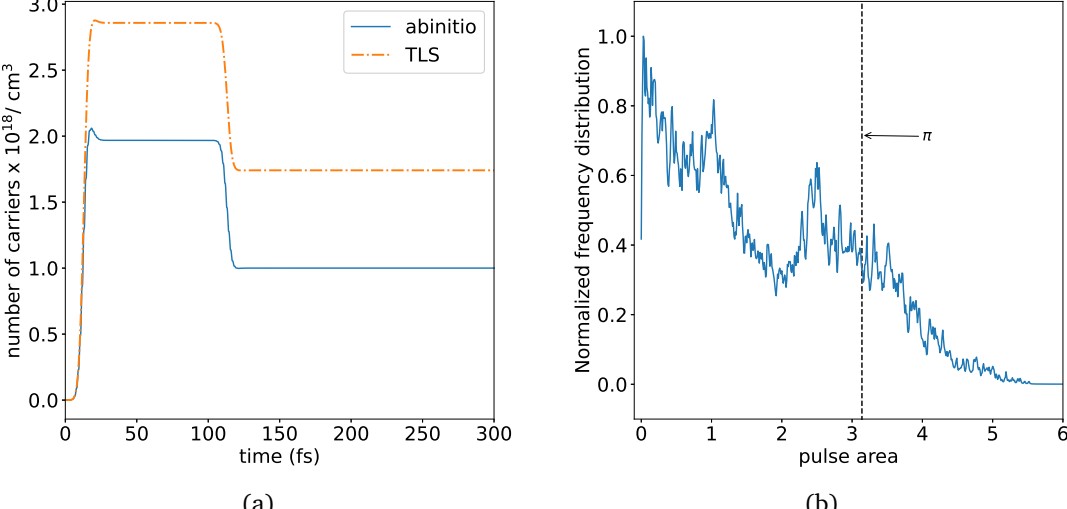

Figure 4: (a) Time dynamics of the carriers in the high-field regime. *ab initio* results (blue continuous line) are compared with TLS simulations (orange dashed line). The two curves evidence the same behavior, whereas the TLS model overestimates the number of the carriers. (b) Normalized frequency distribution of the values of the pulse area for all the transitions activated by the laser pulse. The vertical dashed line indicates the area of the $\pi$-pulse.

We observe that, in optimum agreement with the complete solution presented in Fig. 3b, the leading linear-order terms $p^{(1,0)}$ and $p^{(0,1)}$ have the same shape and magnitude, but $p^{(0,1)}$ is shifted at $t = \tau$. The echo signal is codified in cubic coefficient and, at $t = 2\tau$, the second addend of $p^{(1,2)}$ becomes independent from the detuning $\delta$ giving rise to the observable echo peak when averaged over the ensemble of TLS.

Due to this analysis we propose a simple formula to estimate the efficiency of the echo signal retrieval, based on the ratio of the amplitude of the pump and echo signal. This provides

$$\lambda'_{TLS} = \frac{\text{echo} - \text{peak}}{\text{pulse} - \text{peak}} = e^{-2\eta\tau} \frac{\langle \theta_p \theta_e^2 \rangle}{4 \langle \theta_p \rangle}, \tag{13}$$

where the $\langle \cdot \rangle$ denotes the average over the ensemble. According to this formula we expect that, at constant pump field, the echo efficiency scales as the intensity of the echo field. Equation (13) provides an estimated efficiency of $\lambda'_{TLS} = 2 \times 10^{-3}$, in excellent agreement with the value derived using the curves of Fig. 2.

## 4 Echo dynamics in the high-field regime

We present the analysis of the polarization dynamics and the echo mechanism in the high-field regime. In this particular case the peak of the intensity of the pump and echo fields is equal to $10^7 \, \text{kW/cm}^2$ and the associated fluence is $37.6 \, \mu\text{J/cm}^2$. This value is well below the damage threshold of GaAs, estimated at $10 \, \text{mJ/cm}^2$, for ultra-short pulses with frequency above the band-gap [41]. The non-equilibrium dynamics of the system and the extraction of the echo signal have been investigated using the same procedure adopted in the low-field regime. Computations are performed with values of the delay $\tau$ between the pump and the echo fields ranging from 50 to 250 fs, with a step of 50 fs.

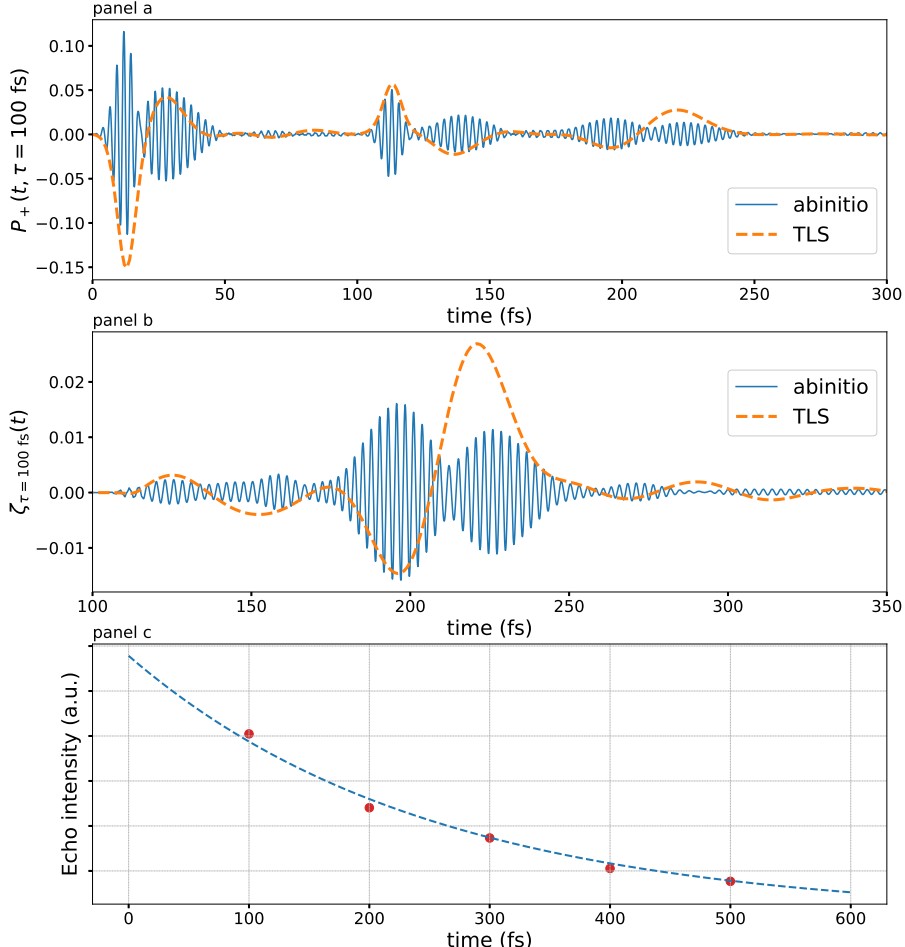

Figure 5: Real-time behavior of polarization and echo signals in the high-fields regime. (a) Time dependent polarization generated by a perturbing field with positive phase $\alpha = 1$ (see Eq. (1)). The *ab initio* result (blue continuous line) is compared with the TLS simulation (orange dashed line). (b) Echo signal defined in Eq. (4). Same convention as panel (a). (c) Time integrated intensity of the $\zeta_\tau$ for different values of $\tau$ (red dots) as a function of $t = 2\tau$ compared with the an exponential function $A e^{-t/T_2}$ (blue dashed line), with dephasing time $T_2 = 250$ fs. A fit is performed to optimize the amplitude $A$. The plot shows that the echo signal time decay is not affected by the FPD mechanism.

The dynamics of the carriers for $\tau = 100$ fs is presented in Fig. 4a. The *ab initio* result shows that the density of the carriers grows up to $2 \times 10^{18}$ electron/cm$^3$ after the pump and the subsequent interaction with the echo field reduces the number of electrons in conduction to around $1 \times 10^{18}$ electron/cm$^3$. A detailed inspection of the occupation levels in function of the band and $\mathbf{k}$-point indexes reveals that the transitions with energy close to $\omega_0$ produce a very strong depletion of one of the valence band in favour of the first conduction state. For some $\mathbf{k}$-points the conduction occupation level almost reach the value of 2, that corresponds to the complete population inversion for bands with spin degeneracy. The subsequent interaction with the echo field produces the opposite behavior with a strong reduction of the excited state occupation level in favour of the valence bands. However, the $\mathbf{K}_{\text{pump}}$ region is a very small volume of the whole BZ. This happens because the conduction band of GaAs at $\Gamma$ is very steep,

and thus the resonance condition with the laser pulse is achieved only close to $\Gamma$. As a consequence, the above discussed population inversion is associated to carriers densities which are commonly considered low in pump and probe experiments in semiconductors. Nevertheless, the laser field intensity corresponds to a strong field regime, and the associated single **k**-point dynamics results from a highly non-perturbative coupling with the perturbing fields.

This behavior is further confirmed by the analysis of the pulse area $\theta$. Since the definition of $\theta$ involves the transition dipoles, we obtain 34875 values, one for each transition. The frequency distribution of the areas is reported in Fig. 4b, and spans a wide range of values, due to the strong dependence of the transition dipoles on **k** in the $\mathbf{K}_{\text{pump}}$ region. The presence of many values around and also well above $\pi$ confirms the non-perturbative regime of the dynamics. Moreover, the wide range implies that standard protocols which are used to maximize the echo efficiency, such as the $\pi/2$ - $\pi$ scheme (here the two values correspond to the area of the perturbing and rephasing field respectively) cannot be realized in this case. Indeed such protocols would require much more narrow distributions of values around the desired one.

The time dependent polarization and echo signal are reported in Fig. 5. The first two panels of the figure show the results corresponding to $\tau = 100\,\text{fs}$. Panel $a$ displays the full polarization signal $P_+$ (blue curve), which represents the non-equilibrium polarization induced by the interaction with the pump and echo field with $\alpha = 1$. We observe two main peaks located at about $t_{P1} = 12\,\text{fs} = t_0$ and $t_{R1} = 112\,\text{fs} = t_1$, in correspondence with the maxima of the pump and of the rephasing fields, respectively. Nonetheless, unlike what happens in the low-field regime, the polarization has a structured time profile, and two more maxima around $t_{P2} = 27\,\text{fs}$ and $t_{R2} = 138\,\text{fs}$ emerge. This fact is an ulterior confirmation that the dynamics for this field intensity is far beyond the linear response regime.

The echo signal $\zeta_\tau(t)$ is plotted in panel ($b$) of Fig. 5. It also evidences a double peak structure with maxima at about $t_{E1} = 196$ and $t_{E2} = 226\,\text{fs}$. This shape is reminiscent of the two peaks $P1$ and $P2$ generated by the pump, however mirrored in time by the echo mechanism. Indeed the time position of the first echo peak $E1$ is consistent with an effective delay *seen* by $P2$ from the rephasing pulse $\tau_2^* = t_1 - t_{P2} = 85\,\text{fs}$, which would imply an echo at $t_{P2} + 2\tau_2^* = 197\,\text{fs}$, in excellent agreement with the observed value of $t_{E1} = 196$. The same analysis applied to the second echo peak foresees that the expected echo would be at $t_{P1} + 2\tau = 212\,\text{fs}$, i.e. slightly before the observed value. This discrepancy may be due to the complex profile time of the polarization.

Lastly, the time integral of the envelope of $\zeta_\tau(t)$ is shown in panel ($c$) as a function of $t = 2\tau$, *i.e.* the value is represented at the time where the echo is expected. The comparison with an exponential function with decay time $T_2 = 250\,\text{fs}$ demonstrates that the intensity of the $\zeta_\tau(t)$ signal scales in time according to $e^{-2\tau/T_2}$, *i.e.* that also in this case the irreversible physical decay time introduced in the EOM for the density matrix can be retrieved. The echo defined according to Eq. (4) provides an FPD-free reconstruction of the original polarization also in the high field regime, where the physical content of the $\zeta_\tau(t)$, as expressed by the perturbative expansion (8), is not limited to the leading cubic contribution measured in FWM experiments. Following the procedure defined around (5), the efficiency of the echo signal retrieval can be computed also in the high field regime, providing the value $\lambda = 0.55$ (while the efficiency $\lambda'$, associated to the ratio of the amplitudes, is equal to 0.3 ). We observe that the efficiency grows of a factor 100 with respect to the low field example and the echo signal becomes clearly observable at the level of the full polarization $P_+$. Indeed, in the time window close to $t = 2\tau$, $P_+(t)$ and $\zeta_\tau(t)$ are very similar, showing that, despite the echo is not realized with a specific $\pi$ protocol (for the $\pi/2$ - $\pi$ protocol $\lambda = 1$), it is anyway an echo in the global signal.

A TLS modelization of the system has been performed also in the high-field regime, using the very same procedure described in the beginning of section 3.2. The results for the carriers,

for the polarization $P_+$ and for echo signal are reported in Fig. 4a, and in the panels ($a$) and ($b$) Fig. 5 (orange dashed lines). A comparison with the *ab initio* results shows that in this regime the TLS is only able to capture the qualitative behavior of the dynamics. In particular the number of carriers is overestimated and the profile of the echo signal is not correctly reproduced. The differences emerge due to the multi-band nature of the optical transition in the real material, that the TLS model is not able to describe. Despite we included all the possible transitions ($v_1 \rightarrow c_1$, $v_2 \rightarrow c_1$, ...), each one is completely uncorrelated in the TLS description while at the *ab initio* level the population of $c_1$ due to the $v_1 \rightarrow c_1$ transition is felt by the $v_2 \rightarrow c_1$ transition and vice versa. The inclusion of such effect in an extension of the TLS would significantly increase the complexity of the modelling. It is thus better to directly work with a full *ab initio* implementation, also in view of further developments.

## 5 Comments and conclusions

We have simulated the free polarization decay and the generation of photon-echo signals in GaAs with an *ab initio* formalism. A detailed convergence study on the computational parameters has been performed, evidencing in particular the important role of the **k**-points sampling in the sector of the BZ which contains the transitions resonant with the energy of the optical pulses. This analysis has shown that a very fine sampling of this region is needed in order to describe the FDP of the polarization signal.

The results of the *ab initio* real-time dynamics have been compared with a simple modelization of the GaAs in which each relevant optical transition is treated as an independent two-levels system. The TLS analysis is carried out in the so called *rotating wave approximation* (RWA) in which only the resonant part of the perturbing field is considered. We have proposed a computational procedure for the extraction of the echo signal which does not rely on the non collinearity of the pump and rephasing pulses, allowing to extract the echo within the dipole approximation, also in the low intensity regime. The TLS results, fed with the energy dispersion and the Rabi couplings of the *ab initio* GaAs, have been used to validate our approach. Perfect agreement between the results of the two approaches is obtained at low intensity. This fact provides also an *a posteriori* justification of the usage of the RWA in the TLS analysis. Instead, in the high intensity regime the TLS modelling shows quantitative deviations from our *ab initio* simulations, mainly due to the fact that the latter are able to go beyond the two states approximation, thus accounting for the multi-band nature of GaAs.

The value of the pulse area $\theta$ has a relevant role in the physics of photon echo mechanisms and has been investigated in our analysis. In standard echo experiments at high intensity, values of $\theta \approx \pi/n$, with $n$ some integer, are used to maximize the echo efficiency. However, as we have verified by looking at the strong dependence of the electronic dipoles on the crystal momentum **k**, the value of $\theta$ cannot be easily controlled when the system is excited in the continuum by an ultra short pulse. Consequently the standard *global echo* procedures which rely on a constant value of the pulse area in the whole optical active **k**-points region are inadequate for systems with this behavior. In doing so we have also shown that, using ultra short laser pulses, the fluences needed to achieve the physics of Rabi flopping are experimentally feasible and well below the the threshold damage in GaAs.

The present work provides a starting point for further developments, both from the experimental and the theoretical point of view. Experimentally, it shows that echo experiment could be performed in semiconductors without the need of extracting the third order response, thus avoiding complex non collinear geometries in the perturbing and rephasing pulses. Theoretically, it opens the way to more refined simulations, beyond the independent particle approximation, where the need of a dense **k**-points sampling in the whole BZ could be accounted for

by means of double grid techniques.

We are presently considering different extensions: (i) to account for intra-band dipoles, which are needed to model laser pulses out of resonance with the optical absorption of the material; (ii) to investigate the effect of electron-hole interaction needed to model echo experiments with laser pulses tuned in resonance with excitonic peaks; and (iii) to account for the update in the effective interaction felt by the electrons due to the non-equilibrium electronic density. This latter point is crucial to verify if the echo signal survives the changes in the band structure, or at which density it would be destroyed.

## A Perturbative expansion scheme of the density matrix

We review the basic equations of the perturbative expansion of the density matrix operator in order to justify the formula (7) for the expansion of the observable polarizations $P_{\pm}$.

This procedure passes through the construction of the chain of perturbative EOM for the density matrix operator

$$\rho_{nm\mathbf{k}}(t) = \rho_{nm\mathbf{k}}^{(0)} e^{-i\omega_{nm\mathbf{k}}t} + \rho_{nm\mathbf{k}}^{(1)}(t) + \rho_{nm\mathbf{k}}^{(2)}(t) + \cdots,$$

where $\rho_{nm\mathbf{k}}^{(0)}$ are boundary condition at $t = 0$ and the frequencies $\omega_{nm\mathbf{k}}$ can contain a complex part to include a dephasing term. The EOM for the $i$-th coefficient in the TD-IP approximation are

$$\partial_t \rho_{nm\mathbf{k}}^{(i)} + i\omega_{nm\mathbf{k}}\rho_{nm\mathbf{k}}^{(i)} = F_{nm\mathbf{k}}^{(i)}[\rho], \tag{14}$$

where the source terms for $i \geq 1$ read

$$F_{nm\mathbf{k}}^{(i)}[\rho] = -i\mathbf{E}(t) \cdot \sum_p \left( \boldsymbol{\mu}_{np\mathbf{k}} \rho_{pm\mathbf{k}}^{(i-1)} - \rho_{np\mathbf{k}}^{(i-1)} \boldsymbol{\mu}_{pm\mathbf{k}} \right). \tag{15}$$

Equations (14) can be hierarchically solved starting from the lowest order. Since $\rho_{nm\mathbf{k}}^{(i)}(0) = 0$ for $i \geq 1$, the formal solution the $i$-th expansion coefficient can be represented as

$$\rho_{nm\mathbf{k}}^{(i)}(t) = e^{-i\omega_{nm\mathbf{k}}t} \int_0^t dt' e^{i\omega_{nm\mathbf{k}}t'} F_{nm\mathbf{k}}^{(i)}[\rho](t'). \tag{16}$$

Thanks to this results we can assess the effect of the phase factor $\alpha$ introduced (1) on the perturbative expansion coefficients in terms of the echo field. Indeed, an inspection of equations (15) and (16) reveals that

$$\rho_{nm\mathbf{k}}^{(i)}(t)|_{\alpha=1} = -\rho_{nm\mathbf{k}}^{(i)}(t)|_{\alpha=-1}, \quad i \in \text{odd},$$
$$\rho_{nm\mathbf{k}}^{(i)}(t)|_{\alpha=1} = \rho_{nm\mathbf{k}}^{(i)}(t)|_{\alpha=-1}, \quad i \in \text{even}, \tag{17}$$

so the even order expansion coefficients are unaffected by the overall change of sign of the echo fields, whereas the odd ones change sign.

## B Two level system based description of optical absorption

We collect some background material concerning the non-equilibrium dynamics induced by an ultrafast optical pulse coupled with a two-level system (TLS).

The unperturbed Hamiltonian of the TLS reads

$$\hat{H}_0 = -\frac{1}{2}\varepsilon\hat{\sigma}_z, \tag{18}$$

where $\hat{\sigma}_z$ is the Pauli matrix and $\varepsilon$ represents the energy gap. The ground state of system is described by a diagonal density matrix with an occupation level equal to 1 for the state with lowest energy.

The coupling with the optical pump is modeled through a dipole interaction, that is

$$\hat{H} = -\hat{\mu} \cdot \mathbf{E}(t), \tag{19}$$

where $\hat{\mu}$ is the dipole operator and we assume that the eigenstates of $\hat{H}_0$ have definite parity, which implies vanishing diagonal matrix elements of the interaction Hamiltonian.

The electric field is parametrized as the perturbing field of (1). Accordingly, the matrix element of $\hat{H}$ responsible for the transition can be expressed as

$$\begin{aligned} H &= -sin(\omega_0 t)\big(\Omega_p f_p(t) + \alpha \Omega_e f_e(t-\tau)\big) \\ &= -sin(\omega_0 t)\Omega(t), \end{aligned} \tag{20}$$

where $\Omega_p = \mu_{12}^x E_p$ and $\Omega_e = \mu_{12}^x E_e$ are the Rabi couplings of the pulse and echo fields (note that we have set $\hbar = 1$ in this analysis) and we have introduced the time-dependent envelope $\Omega(t)$ as a compact notation.

We are interested in probing the system with pulse's energy close to the energy gap, so we introduce the *detuning* $\delta$ as

$$\omega_0 = \varepsilon + \delta \tag{21}$$

and the condition $\delta \ll \varepsilon$ is satisfied. In this case we can adopt the rotating wave approximation (RWA) in which the matrix element of the transition Hamiltonian reduces to

$$H = \frac{i}{2}\Omega(t)e^{i\omega_0 t}. \tag{22}$$

It is convenient to describe the dynamics of the system in terms of *polarization $p$* and of the *inversion $I$* variables defined as

$$p = e^{-i\omega_0 t}\rho_{12}, \quad I = \rho_{11} - \rho_{22}. \tag{23}$$

Note that the oscillating phase factor in the definition of the polarization realizes the so called formulation in the *rotating frame*. In terms of these variables the EOM for the density matrix read

$$\begin{aligned} \dot{p} &= -i\delta p - \frac{1}{2}|\Omega|I - \eta p, \\ \dot{I} &= |\Omega|(p + p^*), \end{aligned} \tag{24}$$

where we have included the term $-\eta p$ that produces a dephasing of the polarization with damping time $T_2 = 1/\eta$. Moreover, without loosing of generality, we have formulated the EOM (24) in terms of the moduli of the envelope function $\Omega(t)$. In general $\Omega(t)$ is complex since it has the same phase of the dipole matrix element $\mu_{12}^x$. However, it is possible to reabsorb this (constant) phase in the definition of the polarization. In the present analysis we assume that this procedure has been performed, thus leading to the equations (24).

Observable quantities, like the number of carriers $n_c$, *i.e.* the fraction of charge in the excited state and the observable polarization $\mathbf{P}$ can be readily computed in terms of $p$ and $I$. In particular, assuming that the (conserved) total charge is equal to 1 the number of carriers is expressed as

$$n_c = \frac{1 - I}{2}, \tag{25}$$

while the ($x$ component) of the polarization is defined as

$$P_x = |\mu_{12}^x|(p + p^*). \tag{26}$$

In the present analysis we have defined an ensemble of N TLS's using the *ab initio* detunings and Rabi couplings, as described in the beginning of section B. Observables are computed as the average over the ensemble of the equations (25) and (26). In view of the quantitative comparison with the results of the *ab initio* calculations the average TLS results have been rescaled by the term $2N/N_k$, where the factor 2 keeps into account the spin degeneration and the ratio $N/N_k$ balance the mismatch between the number of the TLS (that actually represents the number of active transitions) and the number of **k**-points.

**Emergence of the echo signal in the perturbative expansion of the TLS equation**

We construct the leading order solution of the TLS equation that gives rise to an echo retrieval mechanism. To this scope we perform a perturbative expansion of the EOM (24) truncated to the linear and quadratic orders in the pump and in the echo amplitudes, respectively. To keep the analysis as simple as possible, we consider pulses with a square-wave profile, so the envelopes $f_p$ and $f_e$ are window functions of width $w$ and amplitude $E_p$ and $E_e$. Moreover, we assume that $e^{i\delta t} \simeq 1$, for time interval of the order of $w$. This condition is well satisfied for short pulses and small detuning.

The TLS is in its ground state before the pump, so the boundary condition associated to the initial part of the dynamics are $p_0 = 0$ and $I_0 = 1$. The first order solution in the pump amplitude, for times longer than $w$, reads

$$p^{(1,0)}(t) = -\frac{1}{2}\theta_p e^{-i\bar{\delta}t}, \quad I^{(1,0)}(t) = 0, \tag{27}$$

where $\bar{\delta} = \delta - i\eta$ is the *complex detuning* that includes the damping parameter and $\theta_p = \Omega_p w$ is the pump area. We remind that the apex notation $p^{(i,j)}$ indicates the $i$-th and $j$-th orders solution in the pump and echo fields.

The perturbative expansion in the echo amplitude starts at $t = \tau$, when the echo pulse is activated, and the associated boundary conditions are

$$p_0 = -\frac{1}{2}\theta_p e^{-i\bar{\delta}\tau}, \quad I_0 = 1, \tag{28}$$

note that the boundary value of the polarization is not zero and represents the zero-th order term in the echo amplitude expansion.

The first order solutions for the polarization and inversion variables, computed for time values longer than the pulse width, are

$$p^{(0,1)}(t) = -\frac{1}{2}\theta_e e^{-i\bar{\delta}(t-\tau)},$$
$$I^{(1,1)}(t) = -\frac{1}{2}\theta_e \theta_p \left( e^{-i\bar{\delta}\tau} + e^{i\bar{\delta}^*\tau} \right). \tag{29}$$

The first order inversion $I^{(1,1)}$ acts as a source term in the second order polarization that can be written as

$$p^{(1,2)}(t) = \frac{\theta_p \theta_e^2}{8} e^{-\eta t} \left( e^{-i\delta t} + e^{-i\delta(t-2\tau)} \right). \tag{30}$$

We recognize that the second addend of (30) is *independent* from the detuning at $t = 2\tau$ and produces an echo signal in which only the irreversible damping $\eta$ is present. A comparison of formula (30) and (27) allows us to assess the efficiency of the echo retrieval, leading to the equation (13).

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
