# Peer review of "Real-time ab initio description of the photon-echo mechanisms in extended systems: the case study of bulk GaAs"

_SciPost Physics, doi:SciPost Phys. 12, 193 (2022)_

## Round 2 · Referee Report · Anonymous (Referee 1) · 2022-2-24

Strengths

  1. Clearly stated problem
  2. Good methodological approach: the ab-initio simulation provides the "numerical experiment" which is then analyzed, to the extent it is possible, in terms of a two level model.

Weaknesses

None

Report

I recommend this paper for publication in the present form.

---

## Round 2 · Referee Report · Anonymous (Referee 2) · 2022-3-22

Strengths

1 Research question clearly exposed
2 Methodology is well explained
3 Clearly points to new avenues of research

Weaknesses

1 Small typos in the manuscript

Report

I believe that the present manuscript is of great quality and deserving of being publish. The authors have provided a clear and detailed method to study photon echos. The model is good enough to provide qualitative results, and evidences both the strengths and limits of a two level system. It is also encouraging that the authors point to further improvements in the model and possible changes in experimental methods that can be used to study this phenomenon.

Requested changes

A) On Page 3 the authors write: "The equilibrium (quasi)-particle band structure $\varepsilon_{n\mathbf k}$ of GaAs is computed using density functional theory (DFT) plus a scissor operator correction". Could they specify if this is a simple rigid opening of the gap (which I believe to be the case), or if band deformation is taken into account?

B) On the topic of the scissor operator, would the band deformation given by GW change the results? My guess is that this can be easily accounted for by increasing the energy width of the laser field.

Some small typos are present through the manuscript. Some examples are (my proposed changes in spelling are in between brackets): 1) "thanks to the development of ultra-short laser–pulse" (laser pulses); 2) "The grey-shaded area evidence" (evidences);

Other typos are: 1) Figure 1: on the k-point path it should read $\Gamma$, not G;

  • validity: high
  • significance: high
  • originality: high
  • clarity: high
  • formatting: excellent
  • grammar: excellent

Author:  Marco D'Alessandro  on 2022-04-21  [id 2400]

(in reply to Report 2 on 2022-03-22)
Category:
answer to question

We would like to thank the referee for the positive review of the paper.

All the small typos have been corrected. For what concerns the scissor operator
we have used a rigid shift of the empty bands of 0.955 eV. The usage of the GW
band structure, in place of the scissor-opened one, could reasonably result in a small
modification of the (shape and peak-position of the) JDOS represented in the right panel
of the FIG.1.
This fact could determine a small variation of the FPD decay time of the polarization. Eventually,
as suggested by the referee, a modification of the energy width of the pulse can be needed to
ensure that the pulse is able to activate all the transitions needed to produce the same
time dynamics of the polarization observed with the scissor-opened bands.

Anonymous on 2022-05-17  [id 2481]

(in reply to Marco D'Alessandro on 2022-04-21 [id 2400])

I accept the reply by the authors.

---

## Round 2 · Referee Report · Anonymous (Referee 3) · 2022-3-23

Strengths

  1. Provide proof of principle for photon-echo experiments
  2. Provide a formula to extract the photon-echo signal in a general experimental set-up
  3. Compare two theoretical approaches analysing their strengths/shortcomings

Weaknesses

  1. There are a few inaccuracies in the computational details
  2. for the high-field the reason for the discrepancies between ab-initio and TLS could maybe be investigated further

Report

The introduction provides motivations and background for the study. The methodology is generally clearly and precisely outlined. Discussion of results is accurate and forthright. The importance of the work is in providing a proof of principle photon-echo experiments, comparing two theoretical approaches and providing a formula to extract the photon-echo signal in a general experimental set-up. I would recommend publishing the manuscript after the changes are implemented

Requested changes

  1. My understanding is that the RWA has been used within the two-level system model. To my knowledge, this is not a good approximation when ultrashort pulses are used. Can the authors comment on that? Can that be the reason for the disagreement observed for the high field?
  2. If the ab-initio approach is reduced to two bands, should one observe the same discrepancies with the converged simulation as for the TLS? did the author conduct such a test?
  3. In the computational details: a)the pseudopotential details (even in a note), b) a_lat is missing the units c) the value of the scissor should be given d) specify whether spin-orbit coupling has been considered, if not it is a reasonable approximation? e) Which bands have been considered in the dynamics
  4. P.8 the notation P^(1,2) is used but it is introduced only later (after eq. 11)
  5. check hyphenation, e.g. semi-conductor => semiconductor; non equilibrium => non-equilibrium; electron hole =>electron-hole....

  • validity: high
  • significance: high
  • originality: top
  • clarity: high
  • formatting: excellent
  • grammar: good

Author:  Marco D'Alessandro  on 2022-04-21  [id 2401]

(in reply to Report 3 on 2022-03-23)
Category:
answer to question

We attach a pdf file with the detailed answers to the questions raised by the referee.

Attachment:

Answer_report3.pdf

---

## Round 2 · Referee Report · Anonymous (Referee 4) · 2022-3-28

Strengths

1- clarity of exposition 2- results are scientifically sound

Weaknesses

1- calculation is only at the Independent Particle (IP) level

Report

The paper is well written and results are scientifically sound. The comparison of the ab initio calculation with the two-level model is a plus.

Requested changes

1- Eq.(3): how is the dipole operator calculated? with which technique?

  • validity: high
  • significance: high
  • originality: top
  • clarity: high
  • formatting: excellent
  • grammar: excellent

Author:  Marco D'Alessandro  on 2022-04-21  [id 2402]

(in reply to Report 4 on 2022-03-28)
Category:
answer to question

We attach a pdf file with the detailed answers to the questions raised by the referee.

Attachment:

Answer_report4.pdf

---

## Round 3 · Author Response

Hereby, we submit a revised version of the paper.
The suggestions and criticisms of the referees have been addressed and some
(minor) modifications have been performed in the manuscript.

---

## Round 3 · List of Changes

1) The usage of the RWA approximation has been discussed and justified. 2) Some computational details have been added in section 2. 3) The reasons of the discrepancy between TLS and ab initio results have been further investigated. 4) Minor errors have been corrected in the text.

---

## Editorial Decision

published